# Intrusion Detection System in the Advanced Metering Infrastructure: A Cross-Layer Feature-Fusion CNN-LSTM-Based Approach

**DOI:** 10.3390/s21020626

**Published:** 2021-01-18

**Authors:** Ruizhe Yao, Ning Wang, Zhihui Liu, Peng Chen, Xianjun Sheng

**Affiliations:** Faculty of Electronic Information and Electrical Engineering, Dalian University of Technology, Dalian 116024, China; yrzhe@mail.dlut.edu.cn (R.Y.); zhliu0910@mail.dlut.edu.cn (Z.L.); PengC@mail.dlut.edu.cn (P.C.); sxianjun@dlut.edu.cn (X.S.)

**Keywords:** smart grid, advanced metering infrastructure (AMI), intrusion detection system (IDS), convolutional neural networks (CNN), long short-term memory (LSTM)

## Abstract

Among the key components of a smart grid, advanced metering infrastructure (AMI) has become the preferred target for network intrusion due to its bidirectional communication and Internet connection. Intrusion detection systems (IDSs) can monitor abnormal information in the AMI network, so they are an important means by which to solve network intrusion. However, the existing methods exhibit a poor ability to detect intrusions in AMI, because they cannot comprehensively consider the temporal and global characteristics of intrusion information. To solve these problems, an AMI intrusion detection model based on the cross-layer feature fusion of a convolutional neural networks (CNN) and long short-term memory (LSTM) networks is proposed in the present work. The model is composed of CNN and LSTM components connected in the form of a cross-layer; the CNN component recognizes regional features to obtain global features, while the LSTM component obtain periodic features by memory function. The two types of features are aggregated to obtain comprehensive features with multi-domain characteristics, which can more accurately identify intrusion information in AMI. Experiments based on the KDD Cup 99 and NSL-KDD datasets demonstrate that the proposed cross-layer feature-fusion CNN-LSTM model is superior to other existing methods.

## 1. Introduction

As the core component of the smart grid, advanced metering infrastructure (AMI) has been strongly developed in recent years [1,2,3]. It uses a bidirectional communication network to connect power companies and customers, collect user consumption data and other information, and implement necessary control measures [4]. However, bidirectional communication networks also provide new avenues for network intrusion; attackers can more easily tamper with or intrude meters through these networks [5]. Tampering with meter readings will cause major economic losses to power companies, and intrusion into the meter will cause the loss of user privacy information, which will seriously affect people’s lives.

In relevant studies, intrusion detection systems (IDSs) have developed as an important means by which to protect the communication security of AMI, and can dynamically detect any offending entity and trigger an alarm [6]. IDSs can be divided into misuse detection and anomaly detection according to the detection technology [7]. Misuse detection mainly identifies attacks by matching features or rules, but it cannot detect unknown attacks. Anomaly detection is a behavior-based detection, which first defines the behavior of the subject’s normal activities and then determines whether the actual behavior of the subject deviates from the normal activities. Therefore, anomaly detection is more suitable for the complex communication environment in AMI.

With the application of artificial intelligence, anomaly detection methods based on artificial intelligence have become a hotspot in IDS research. As an important branch of artificial intelligence, traditional machine learning (ML) has been applied in AMI intrusion detection due to its advantages including automatic feature extraction, independence from prior knowledge, and easy design and construction [8,9,10,11]. However, intrusion detection methods based on ML cannot process large amounts of nonlinear high-dimensional data, so they are difficult to adapt to the increasingly complex and diversified attack environment of AMI. Deep learning (DL) is a subclass of ML. It uses a deep neural network to express features and perfectly solves the defects in ML, such as its low accuracy, inability to process complex data, and poor classification effect, and has therefore been gradually applied in AMI intrusion detection research [12,13,14,15].

Considering the periodic, high-traffic, and nonlinear characteristics of AMI communication, a new DL model for AMI intrusion detection is proposed in the present work. The proposed model is composed of a combination of convolutional neural network (CNN) and long short-term memory (LSTM) components via cross-layer feature-fusion. The CNN component can extract global features, while the LSTM component can extract periodical features. After the two types of features are fused via a feature fusion component, multi-scale and multi-domain abnormal information can be detected. This model combines the advantages of both the CNN and LSTM, so it exhibits good performance in AMI intrusion detection. 

The main contributions of this paper are as follows:A cross-layer feature-fusion CNN-LSTM intrusion detection model is proposed. Compared with other models, the proposed model combines the characteristics of the CNN and LSTM and can more effectively identify intrusion information in AMI;The fusion feature is adopted to represent the multi-domain characteristics of the data. This avoids the limitations of single features and achieves the complementation of advantages among different features;The proposed model was evaluated on the KDD Cup 99 and NSL-KDD datasets, both of which are rich in samples and contain all possible types of attacks of AMI. The Experimental results demonstrate that the proposed cross-layer feature-fusion CNN-LSTM intrusion detection model exhibits better performance than traditional intrusion detection models.

The remainder of this paper is arranged as follows. Section 2 summarizes the relevant studies. Section 3 introduces the system components of the cross-layer feature-fusion CNN-LSTM intrusion detection model. Section 4 describes the analysis and preprocessing of the dataset. Finally, Section 5 presents the experimental process and results.

## 2. Related Work

The concept of intrusion detection has been widely implemented since it was first put forward by Anderson [16]. Most initial intrusion detection methods in the field of AMI intrusion detection used statistical techniques [17,18]. However, with the development of artificial intelligence technology, increasingly more ML and DL methods have been applied.

### 2.1. AMI Intrusion Detection Based on Traditional Machine Learning

ML can be divided into supervised learning and unsupervised learning. Most ML models are shallow-layer models with simple structures and strong generalization ability, so they are widely used in AMI intrusion detection.

Jokar et al. [19] put forward a detection model of electricity theft based on consumption patterns. This model applies ML classification and clustering technology, and was combined with a transformer instrument to monitor customers with abnormal electricity consumption. It was found to exhibit high accuracy and maintain strong robustness. Additionally, a real-time distributed intrusion detection system (DIDS) suitable for AMI was proposed by Alseiari et al. [20]. This model uses unsupervised online small-batch *k*-means clustering technology to monitor the data flow in AMI. Vijayanand et al. [21] used an IDS constructed with a multi-support vector machine (SVM) classifier to conduct the early detection of threats in AMI. Each classifier detects only a specific attack, which solves the problem of the weak multi-classification ability of SVMs. A power theft detection scheme based on a decision tree (DT) and SVM was proposed by Jindal et al. [22]. The combined DT-SVM classifier is capable of accurately detecting real-time power theft at all levels of power transmission and distribution. Li et al. [23] proposed an AMI intrusion detection model based on the online sequence extreme learning machine (OS-ELM). This model can utilize online sequence training and achieves a faster detection speed while ensuring accuracy. To address the “black hole attack” of AMI, Boumkheld et al. [24] proposed an intrusion detection model based on a naive Bayesian network that can effectively solve this challenge. Jokar et al. [25] proposed a Zigbee-based intrusion detection model for AMI. The model uses an intrusion detection system based on Q-learning to protect the network from attacks and learns the best strategy to deal with attacks by interacting with the environment. A two-level network intrusion protection system for AMI was proposed by Almakrami et al. [26]. In the phase of intrusion detection, the SVM is used as the detection algorithm to find suspicious events in AMI. Khan et al. [27] proposed a hybrid intrusion detection model, which balances the dataset with an improved *K*-nearest neighbor (KNN) algorithm. The Bloom filter is then used to detect abnormal data within the AMI system.

However, ML methods generally exhibit disadvantages including easy overfitting, poor performance on multi-classification tasks, and low accuracy. These disadvantages limit the use of ML for intrusion detection in AMI.

### 2.2. AMI Intrusion Detection Based on Traditional Deep Learning

DL is composed of feature extraction components and deep neural networks. It can directly learn features from a large amount of data, does not rely on feature engineering, and has an excellent multi-classification ability. Therefore, DL is becoming a research hotspot in the field of AMI intrusion detection.

In the research by He et al. [28], the conditional deep belief network (DBN) was adopted to identify false injection attacks in a smart grid. This detection model requires few external conditions and can achieve high accuracy. A wide and deep CNN model for the detection of electricity theft was proposed by Zheng et al. [29]. The model consists of a deep component and a wide component, which are, respectively, used to identify the periodicity of electricity consumption and capture the global characteristics of electricity consumption data. Ullah et al. [30] proposed a hybrid deep neural network (HDNN) intrusion detection model by combining the CNN, the gated recursive unit (GRU), and the particle swarm optimization (PSO) algorithm. The model uses the CNN for feature selection and extraction, and GRU-PSO technology is used to classify the provided data. The system can automatically perform the processes of feature extraction and classification. In the research by Liu et al. [31], a CNN was used to identify an intrusion, and the accuracy of the model was improved via data enhancement technology. Xiao et al. [32] adopted an auto-encoder (AE) to reduce the dimension of the data to decrease the interference of redundant features, and a CNN was adopted to identify the intrusion information. In the research by Yang et al. [33], an improved CNN was adopted to identify intrusion information. The CNN is improved to extract features across layers, and feature fusion is used to obtain comprehensive features. Shen et al. [34] applied an extreme learning machine (ELM) to intrusion detection, which was found to improve the detection speed and generalization ability of the model. In the research by Zhang et al. [35], a smart grid intrusion detection model that combines the genetic algorithm (GA) and ELM was proposed. The model retains the advantages of the ELM, and the GA is introduced to ensure the optimal parameters of the model. Staudemeyer et al. [36,37] introduced LSTM into the field of intrusion detection, explored the correlation of the temporal domain of intrusion information, and effectively reduced the rate of false positives. A bidirectional GRU (BiGRU) was used in the research of Xu et al. [38] to detect abnormal data. Compared with LSTM, the BiGRU is more efficient and exhibits higher accuracy and lower false-positive rates. Hasan et al. [39] combined a CNN and LSTM, and proposed a serial CNN-LSTM electricity-stealing detection model to simultaneously consider the local and periodic characteristics of electricity information. A hierarchical spatiotemporal features-based intrusion detection system was proposed by Wang et al. [40]. It uses use CNN and LSTM to simultaneously learn the low- and high-level spatiotemporal characteristics of packet bytes to complete intrusion detection. Vinayakumar et al. [41] connected CNN and LSTM and showed a serial CNN-LSTM intrusion detection system model to extract high level feature representations that represents the abstract form of low level feature sets of network traffic connections.

Compared with ML intrusion detection, single DL models have achieved significant progress in accuracy improvement, multi-classification ability enhancement, and overfitting reduction, but they still have disadvantages, such as single identification features, their ease of falling into the local optimum, and their slow convergence speed. While serial CNN-LSTM combines the advantages of different models, the input of the LSTM network is processed by the CNN, which will lead to the loss of some periodic features; thus, the effect is not ideal. To solve these problems, this paper proposes an intrusion detection model based on the cross-layer feature fusion of CNN and LSTM components. The model utilizes CNN and LSTM components to respectively extract global and periodic features, and exhibits better intrusion detection performance. In addition, the depth of the cross-layer feature-fusion CNN-LSTM neural network is shallower, which can effectively avoid the problem of gradient disappearance during the backpropagation process.

## 3. System Components

In AMI, the characteristic distribution of normal electricity information is very regular [35] and has obvious periodicity [29], while abnormal electricity information does not have these characteristics. Therefore, this paper proposes a cross-layer feature-fusion CNN-LSTM intrusion detection model, the architecture of which is illustrated in Figure 1. The model is mainly composed of data preprocessing, CNN, LSTM, and feature fusion components. In the data preprocessing component, the input is numerically processed and normalized to meet the requirements of the neural network. The CNN component consists of convolutional layers, pooling layers, and fully connected (FC) layers, and its main function is to extract local features and detect whether the feature distribution of electricity consumption information is normal. The LSTM component is composed of several LSTM cells, and it is mainly used to detect the periodicity of electricity information via its memory function. The feature fusion component is composed of multi-layer perceptrons (MLPs), which are mainly used to fuse the features extracted from the CNN and LSTM components and to normalize the classification probability to obtain the final result.

### 3.1. Convolutional Neural Networks Component

The CNN is composed of five parts, namely an input layer, convolutional layer, pooling layer, FC layer, and output layer [42]. Different CNNs have different layer configurations. The structure of the CNN used in the present study is shown in Figure 2, and is composed of an input layer, four convolutional layers, two pooling layers, and two FC layers.

The function of the convolutional layer is to extract features from the data. It contains multiple layers of convolution kernels, each of which corresponds to a weight and a deviation coefficient. When the convolution kernel *i* is in operation, the weight coefficient is assumed to be *w_i_*, the deviation quantity is *b_i_*, and the input of convolutional layer *i* is *X_i_*_−1_. The convolution process can be expressed as:(1)$$X_{i}=f(w_{i}⊗X_{i-1}+b_{i}),$$
where *X_i_* is the output result of convolution kernel *i*, ⊗ represents the convolution operation, and *f*(*x*) represents the activation function. 

The convolution kernel regularly sweeps the input data to extract the characteristic information. Additionally, *ReLU* is adopted as the activation function of the convolutional layer. Compared with the *sigmoid*, *tanh*, and other activation functions, the derivation of the *ReLU* activation function is easier, which can speed up model training and effectively prevent gradient disappearance. *ReLU* can be expressed as:(2)$$ReLU(X_{i})\;=\{\begin{array}{c} X_{i}(X_{i}>0)\\ 0(X_{i}\leq 0) \end{array},$$

The main function of the pooling layer is to realize invariance and reduce the complexity of the CNN by eliminating redundant information via downsampling. There are two main ways to complete pooling, namely average pooling and max pooling. Average pooling means that the average value in the calculation area is taken as the pooling result of the area, while max pooling means that the maximum value in the area is chosen as the pooling result. Compared with average pooling, max pooling can retain more critical information; therefore, the max pooling method is adopted in the present study. Max pooling can be represented as:(3)$$Q_{j}=Max(P_{j}^{0},P_{j}^{1},P_{j}^{2},P_{j}^{3}\ldots P_{j}^{t}),$$
where *Q_j_* represents the output result of the pooling region *j*, *Max* is the max pooling operation, and *P^t^_j_* is the element *t* of the pooling region *j*.

FC layers act as “classifiers” in the entire CNN. Their main function is to weight the features of the convolutional and pooling layers mapped to the hidden-layer space, and re-map them to the sample-marker space. In the FC layer, a corresponding dropout operation is set up to randomly discard neurons to prevent the occurrence of over-fitting.

### 3.2. Long Short-Term Memory Networks Component

The recurrent neural network (RNN) is the most famous model for training temporal data, but the traditional RNN is difficult to train due to gradient explosion or disappearance. To solve these problems, LSTM uses units with a memory function to replace the hidden units in the RNN [43]. LSTM has long-term memory due to its slow weight changes over time, and can also activate short-term memory in a short-range form. The LSTM structure used in the present study is shown in Figure 3. The core information of the LSTM is transmitted along the horizontal line, and the LSTM forgets the old information and learns the new information via the three gate structures of the forget gate, input gate, and output gate.

The forget gate determines how much information is forgotten. Its inputs are *h_t−_*_1_ and *x_t_*, and the gate outputs a number in the interval [0,1] to the current cell state *C_t−_*_1_; 1 indicates “completely retained”, and 0 indicates “completely discarded”. The specific expression of the forget gate can be expressed as follows:*f_t_* = *σ*(*W_f_*∙[*h_t_*_−1_,*x_t_*] + *b_f_*),(4)
where *h_t_*_–1_ represents the output of the previous cell, *x_t_* is the input of the current cell, *σ* is the *sigmoid* function, and *W_f_* and *b_f_* are the weight and bias, respectively.

The input gate determines the update status of the information and consists of two main steps:(1)The *sigmoid* function is used to determine which contents need to be updated;(2)The *tanh* function is used to generate alternative contents for updating.

Finally, the results of these two steps are combined to update the cell state. The expression forms of these two steps are, respectively, as follows:*it = σ(Wi∙[h_t_*_−1_*,x_t_] + bi),*(5)
(6)$$\tilde{C}t\;=\;tanh(Wc\cdot [h_{t-1},x_{t}]\;+\;bc).$$

When the cell state is updated from *C_t_*_−1_ to *C_t_*, the relevant information must first be discarded and then combined with *i_t_* × $\tilde{C}$ to generate a new cell state. The specific expression is as follows:(7)$$C_{t}\;=\;f_{t}\;\times \;C_{t-1}\;+\;i_{t}\;\times \;{\tilde{C}}_{t},$$

The output gate determines the final output, which is based on the current cell state, and is also an input for the next cell state. It consists of two main steps:(1)The *sigmoid* function is used to determine what content will be output;(2)The *tanh* function is used to propose the cell state and obtain the final output of the output gate.

The specific expressions of the output gate are as follows:*ot = σ(Wo[h_t_*_−1_*,x_t_] + b_o_),*(8)
*ht = ot × tanh(C_t_).*(9)

### 3.3. Feature Fusion Component

The feature-fusion component used in this study is the MLP structure. As shown in Figure 4, an MLP is usually composed of an input layer, output layer, and several hidden layers. A full connection is adopted between adjacent layers, and a corresponding activation function is set to realize nonlinearity. The features extracted by the CNN and LSTM components are combined into comprehensive features with multi-domain characteristics after flattening treatment and contact operation. The comprehensive features enter the MLP through the input layer, and nonlinear mapping is performed in the hidden layer. Finally, the output layer outputs the predicted classification results. After comparison with the real results, the parameters of each layer are corrected via the back propagation of loss, and model training is completed after several parameter corrections.

### 3.4. Model Training

Model training primarily includes the two processes of forward propagation and back propagation, which are completed by the following three steps.

(1)Data preprocessing and two-dimensional mapping. First, the input is numerically processed and normalized to facilitate CNN and LSTM processing. The preprocessing results meet the requirements of LSTM. However, the input form of the CNN in this work is a two-dimensional structure, so the standardized data are processed by two-dimensional mapping. Finally, the data are input into the CNN and LSTM components. The specific process is described in detail in Section 4.(2)Feature extraction and fusion. The features are, respectively, extracted by the CNN and LSTM components, and the fusion of global and periodic features is completed by the feature fusion component. High-dimensional mapping is then completed in the hidden layer of the MLP, and the *softmax* classifier is used to identify different intrusion behaviors. The *softmax* classification model is the extension of the logistic regression model in a multi-classification problem, and maps the output of multiple neurons to the interval (0,1). The equation is given by Equation (10), where *z* represents the input of the *softmax* layer and *C* represents the input dimension. (10)$$S{\left(z\right)}_{i}=\frac{e^{z_{i}}}{\sum_{j=1}^{C}e^{z_{j}}},i=1,\ldots ,C,$$
(3)Backpropagation and parameter updating. After classification by *softmax*, the cross-entropy loss function is first used to calculate the loss between the predicted and actual values. The cross-entropy loss function is given as follows:(11)$$H(p,q)=-\sum_{i=1}^{n}p(x_{i})\log(q(x_{i})),$$
where *p*(*x*_i_) and *q*(*x_i_*), respectively, represent the real and predicted distributions of sample *i*, and *H* represents the final loss value. Back-propagation is then carried out according to the loss value. The Adam optimizer is adopted for the back-propagation process to update the weight and bias of each layer.

## 4. Dataset Selection and Preprocessing

### 4.1. Dataset Selection

AMI is composed of a wide area network (WAN), home area network (HAN), and neighborhood area network (NAN), which are connected with household appliances, smart electricity meters, concentrators, data processing centers, and other critical nodes. Wired or wireless communication is adopted for bidirectional communication between the equipment. From the perspective of the communication rate, the internal communication method of the HAN is mainly low-speed and short-distance communication, and the intelligent appliances in the HAN are connected to the Internet, and are therefore more vulnerable to denial-of-service (DOS) and probe attacks. The NAN also adopts low-speed and short-distance communication. Its internal collection of data is uploaded by the HAN, and it is characterized by low computing and storage capabilities; therefore, it cannot effectively defend against intrusions, so it is more vulnerable to user-to-root (U2R) attacks. The WAN is mainly based on high-speed and long-distance communication, and the data it transmits are sensitive and private. Once the WAN is attacked, the operation of the power grid will be seriously affected, so it is more vulnerable to remote-to-local (R2L) attacks.

The KDD Cup 99 dataset is widely used in the IDS field [44]; it is rich in samples, and includes 4,898,431 pieces of data. The NSL-KDD dataset is an improved version of the KDD Cup 99 dataset in which redundant data were removed, making the distribution of the dataset more balanced and reasonable [45,46]. Although some of the features in these two datasets are character-based features that cannot be processed by the deep learning model, and although some features differ too much or are not conducive to the final classification of intrusion detection, numerical, normalized, and feature-screening processing can alleviate these problems. In addition, these two datasets include the four categories of DOS, probe, U2R, and R2L attacks and 39 attack subclasses, covering all possible attack types of the AMI, and the data distribution has corresponding periodic characteristics. Considering the comparability of experiments, 10% of the training data was selected from the KDD Cup 99 and NSL-KDD datasets as AMI intrusion detection datasets, and the training set and test set were divided according to the 10-fold cross-validation method. The distribution results are shown in Table 1 and Table 2.

### 4.2. Dataset Preprocessing

Each piece of data in these datasets contains 42 features, 38 of which are numerical features, three of which are symbolic features, and one of which is a label. However, the CNN and LSTM are unable to process symbolic features, and the CNN in this paper is a two-dimensional structure; therefore, the data in the dataset must first be preprocessed, mainly via numerical, one-hot, normalization, and dimension reduction processing.

#### 4.2.1. Numerical and One-Hot Processing

During data preprocessing, the function of numerical and one-hot processing is to transform the symbolic features into numerical eigenvectors. The main features that require numerical and one-hot processing are the *protocol_type*, *service*, and *flag* features in these datasets. *Protocol_type* has three attributes, namely the transmission control protocol (TCP), user datagram protocol (UDP), and Internet control message protocol (ICMP). After numerical and one-hot processing, they can respectively be represented by the 1 × 3 dimensions vectors (0,0,1), (0,1,0), and (1,0,0). Similarly, *service* and *flag* contain 70 and 11 attributes, respectively, which can respectively be represented by 1 × 70 dimension and 1 × 11 dimensions vectors.

After numerical and one-hot processing, these three symbolic features are mapped to 1 × 84 dimensions numerical features, which are combined with the original 1 × 38 dimensions numerical features to ultimately obtain 1 × 122 dimensions numerical features.

In addition, the labels of the symbol type must also be numerically processed. There are five types of labels in these datasets, which respectively represent one type of normal behavior (Normal) and four types of attacks (DOS, Probing, R2L, and U2R). The label numerical processing results are presented in Table 3.

#### 4.2.2. Normalization

After numerical and one-hot processing, the high numerical characteristics will be amplified if the original value is used directly. To eliminate the influence of excessively large feature differences, the features in the datasets must be normalized and mapped uniformly in the interval [0,1]. The specific normalization equation can be expressed as follows:(12)$$x=\frac{X-MIN}{MAX-MIN},$$
where *X* is the original eigenvalue, *MIN* represents the minimum value of the feature, *MAX* represents the maximum value of the feature, and *x* is the normalized result.

#### 4.2.3. Dimension Reduction

The data input into the CNN have a two-dimensional structure, while the numerical results have 1 × 122 dimensions, which is a one-dimensional structure. Daweri et al. [47] pointed out that a portion of the features in the KDDCup 99 dataset are redundant, which is not conducive to the realization of classification. Therefore, the discrete cuttlefish algorithm (D-CFA) is used to carry out feature screening to retain the most effective 1 × 100 dimension features. After that, the reserved features are mapped into a 10 × 10 matrix to adapt to CNN processing.

## 5. Experiments and Results

### 5.1. Experimental Environment and Hyper-Parameter Setting

In this research, the training and testing of the model were completed in the Windows operating system, and TensorFlow in the Python DL library was used to realize the programming of the proposed cross-layer feature fusion CNN-LSTM intrusion detection model. The specific hardware and software configurations are reported in Table 4.

Regarding the parameter settings, the learning rate was set as 0.007, and the dropout rate was set as 50%, i.e., half of the neurons were randomly discarded. Additionally, the number of experimental epochs was 100. All the hyper-parameter settings are presented in Table 5.

### 5.2. Evaluation Metrics

In the intrusion detection process, metrics such as the accuracy (*ACC*), precision (*P*), detection rate (*DR*), *F*-measure (*F*), and false-positive rate (*FPR*) are usually used to evaluate the effect of the model [48]. *ACC* refers to the ratio of the number of correctly classified samples to the total number of samples. When the dataset is balanced, *ACC* is an appropriate indicator. Because the dataset used in this research is unbalanced, *ACC* is used as an auxiliary index to evaluate the performance, and its specific calculation is as follows:(13)$$ACC=\frac{TP+TN}{TP+FP+TN+FN},$$

In the Equation (13), *TP* (true positive) represents the number of correctly identified abnormal samples, *FP* (false positive) represents the number of incorrectly identified abnormal samples, *TN* (true negative) represents the number of correctly identified normal samples, and *FN* (false negative) represents the number of incorrectly identified normal samples.

*P* is defined as the ratio between the number of correctly identified abnormal samples and the true number of abnormal samples. It represents the confidence of attack detection, and its specific calculation is expressed as follows:(14)$$P=\frac{TP}{TP+FP},$$

*DR* is defined as the ratio between the number of correctly identified abnormal samples and the predicted number of abnormal samples. The *DR* reflects the ability of the model to identify attacks, which is an important indicator in IDSs. Its specific calculation is expressed as follows:(15)$$DR=\frac{TP}{TP+FN},$$

*F* is defined as the average harmonic value of *P* and *DR*, and its specific calculation is expressed as follows:(16)$$F=\frac{2*P*DR}{P+DR},$$

*FPR* is defined as the ratio of the number of misidentified abnormal samples to the predicted number of normal samples. Its equation is given as follows:(17)$$FPR=\frac{FP}{TN+FP},$$

To comprehensively evaluate a model, multiple metrics are often used simultaneously in intrusion detection research. In this work, *ACC*, *P*, *F*, *DR*, and *FPR* were selected to evaluate the performance of the proposed cross-layer feature fusion CNN-LSTM intrusion detection model in multiple experiments.

### 5.3. Experimental Design and Results

To evaluate the performance of the model, the following three groups of experiments were conducted on the KDD Cup 99 and NSL-KDD datasets.

Experiment 1: The datasets were used to train the cross-layer feature fusion CNN-LSTM intrusion detection model, its convergence ability and classification ability of different attack types (DOS, probe, R2L, U2R) were tested, and its *ACC*, *DR*, and *FPR* values were calculated.

Figure 5 presents the relationship between the training loss and the number of epochs of the proposed cross-layer feature fusion CNN-LSTM model. It can be seen from the figure that with the increase of the number of epochs, the training loss gradually decreased and became stable in the 10th epoch. This demonstrates that the structural design and hyperparameter settings of the model are reasonable, and that it exhibited good convergence ability. The trained model was tested, and the obtained confusion matrices are shown in Figure 6a,b.

The evaluation parameters of the four types of attacks obtained from the confusion matrices are shown in Figure 7a,b. On the KDD Cup 99 and NSL-KDD datasets, the *ACC* values of the model were, respectively, 99.95% and 99.79%, and the *P* values of the DOS and probe attacks were also greater than 99%. However, the detection capabilities of the model for U2R were 71.43% and 50%, respectively, mainly because the number of training samples of U2R was limited, accounting for only 0.01% and 0.04% of the entire datasets, respectively.

To verify the experimental results, a five-fold cross-validation experiment was also carried out for the proposed model, and the experimental confusion matrixes are shown in Figure 8. The *P*, *DR*, and *FPR* values of the five communication types in these two datasets were obtained from the confusion matrixes, as shown in Figure 9a,b. In the five-fold cross-validation experiment, the *P* values of the proposed model for the three communication types of Normal, DOS, and Probing were greater than 99%, which is consistent with the previous experimental results.

The results of Experiment 1 demonstrate that the proposed cross-layer feature fusion CNN-LSTM intrusion detection model exhibited good performance in terms of *P*, *DR*, and *FPR* for different types of attacks.

Experiment 2: The cross-layer feature fusion CNN-LSTM, CNN, LSTM, NLP, and serial CNN-LSTM models were, respectively, trained and tested, the *DR* and *F* measures of these five models were determined, and the performance improvement effect of the proposed model was compared.

In Experiment 2, the *DR* and *F* measure were selected to evaluate the performance improvement of the proposed cross-layer feature fusion CNN-LSTM model. The two datasets were used to train and test the CNN, LSTM, MLP, and serial CNN-LSTM models, respectively, and the test results were compared with those of the proposed model. Figure 10, Figure 11, Figure 12 and Figure 13, respectively, present the *DR* and *F* measure of the different models on the two datasets. Taking the KDD Cup 99 dataset as an example, compared with the best performance of the serial CNN-LSTM, the detection performances of the proposed model for the four types of attacks were improved, especially for the detection of U2R; the *DR* value of the proposed model was increased from 37.50% to 71.43%, and the *F* measure was increased from 50% to 76.92%.

The results of Experiment 2 demonstrate that the cross-layer feature fusion resulted in obvious improvements compared to the single models and serial CNN-LSTM, and effectively avoided the limitation of extracting features from a single model and the feature loss of serial CNN-LSTM. This result was also proven on different datasets.

Experiment 3: The proposed model was compared with methods proposed in previous studies. The comparison results are shown in Table 6. Compared with the previous methods, the performance of the proposed cross-layer-aggregated CNN-LSTM intrusion detection model was improved in terms of *ACC*, *DR*, and *FPR*. However, these results are not absolute, as the randomness of the training set and test set selection may have led to different final results. Nevertheless, it is believed that the proposed model is better than existing models proposed in previous studies.

## 6. Conclusions

In this paper, an intrusion detection model based on cross-layer feature-fusion CNN-LSTM was proposed for the detection of intrusions in AMI. The model is composed of CNN and LSTM components, which can respectively detect the local and periodic characteristics of electricity data. The KDD Cup 99 and NSL-KDD datasets were used to train and test the model. The experimental results demonstrated that the performance of the proposed model was superior to those of a single DL component and models proposed in previous studies, and can therefore guarantee AMI communication security. Future research should mainly focus on the following aspects. (1) Due to the limited number of samples in the datasets, the detection effect of the U2R attack was not ideal; thus, research on small-sample intrusion detection should be considered. (2) There remains a certain difference between experimental datasets and real AMI communication data, so a real AMI communication dataset should be constructed, and the model should be improved according to the experimental results to better meet the requirements of AMI intrusion detection.

## Figures and Tables

**Figure 1 sensors-21-00626-f001:**
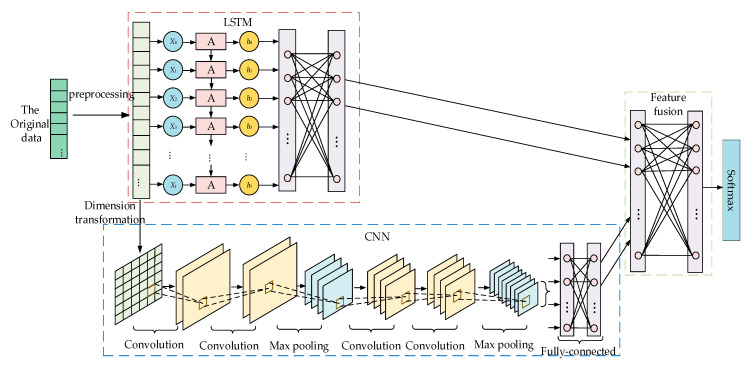
Cross-layer feature-fusion CNN-LSTM intrusion detection model.

**Figure 2 sensors-21-00626-f002:**
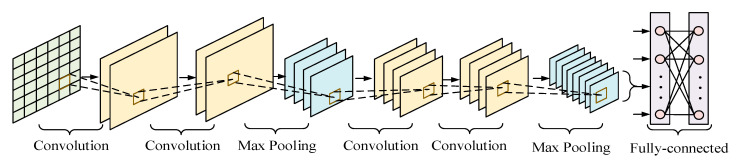
Convolutional neural networks architecture.

**Figure 3 sensors-21-00626-f003:**
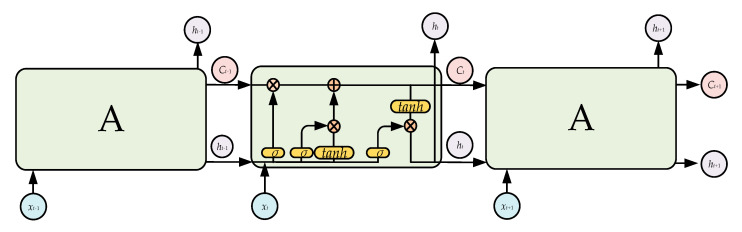
Long short-term memory architecture.

**Figure 4 sensors-21-00626-f004:**
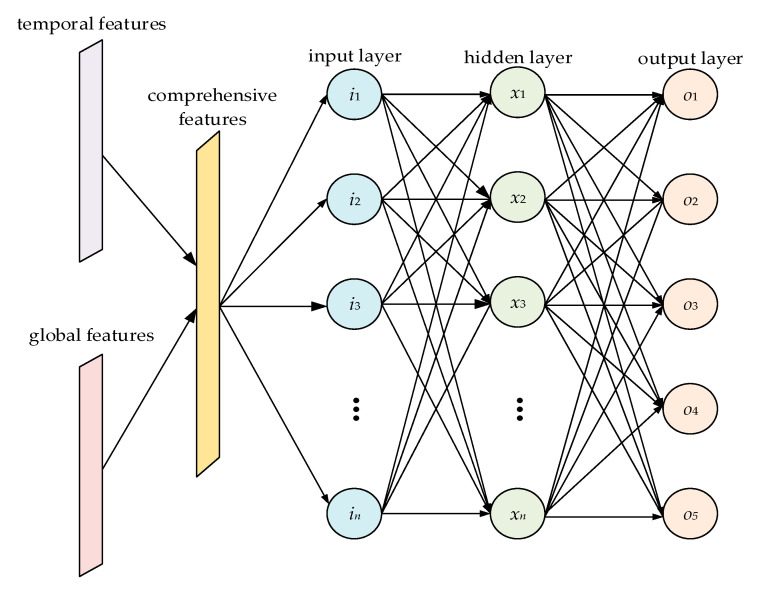
Feature fusion component architecture.

**Figure 5 sensors-21-00626-f005:**
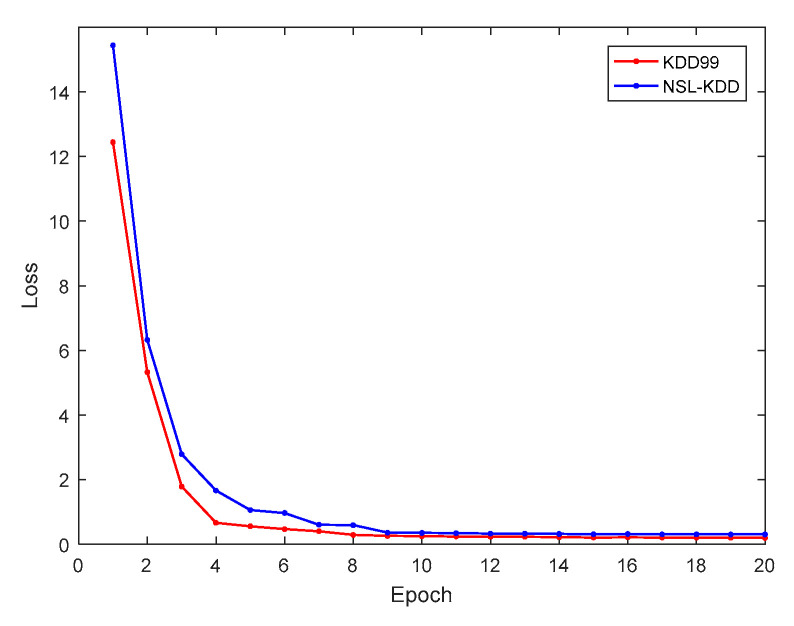
Relationship between training loss and epochs.

**Figure 6 sensors-21-00626-f006:**
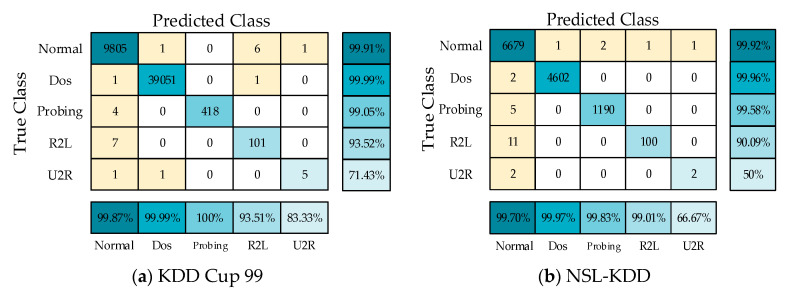
Confusion matrix on KDD Cup 99 and NSL-KDD.

**Figure 7 sensors-21-00626-f007:**
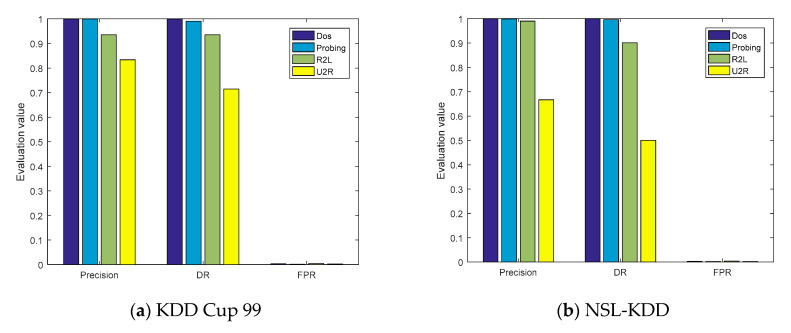
The evaluation parameters of cross-layer feature fusion CNN-LSTM.

**Figure 8 sensors-21-00626-f008:**
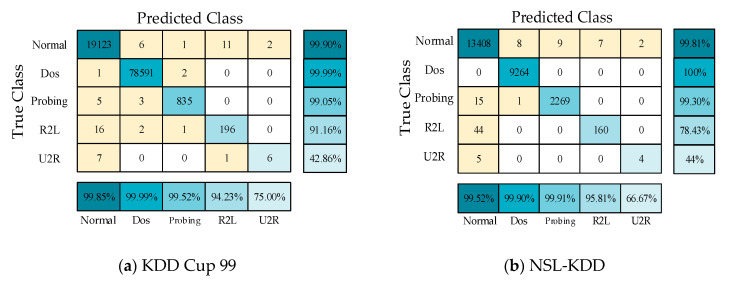
Confusion matrixes on the KDD Cup 99 and NSL-KDD datasets in the five-fold cross-validation experiment.

**Figure 9 sensors-21-00626-f009:**
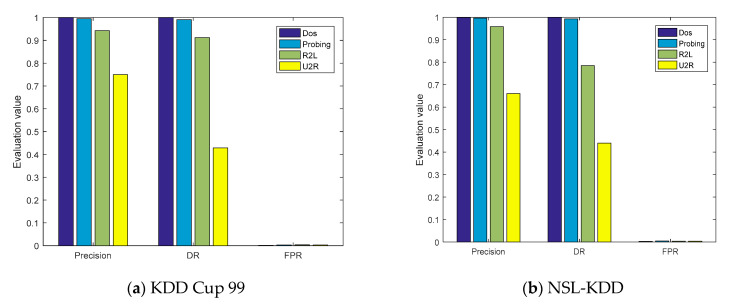
The evaluation parameters of the cross-layer feature-fusion CNN-LSTM model in the five-fold cross-validation experiment.

**Figure 10 sensors-21-00626-f010:**
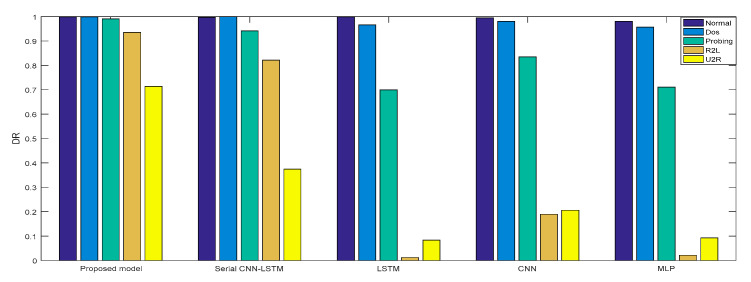
*DR* of KDD Cup 99.

**Figure 11 sensors-21-00626-f011:**
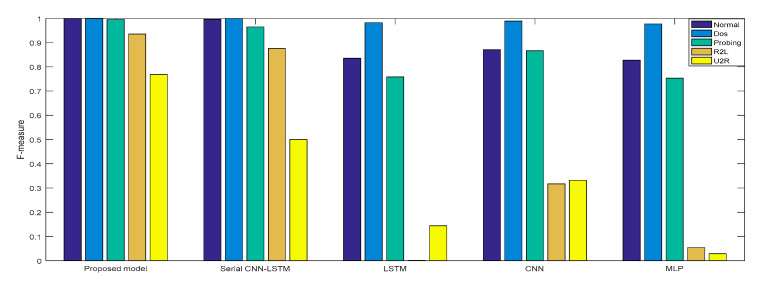
*F*-measure of KDD Cup 99.

**Figure 12 sensors-21-00626-f012:**
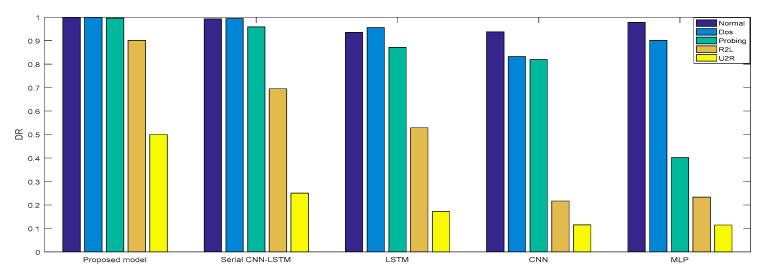
*DR* of NSL-KDD.

**Figure 13 sensors-21-00626-f013:**
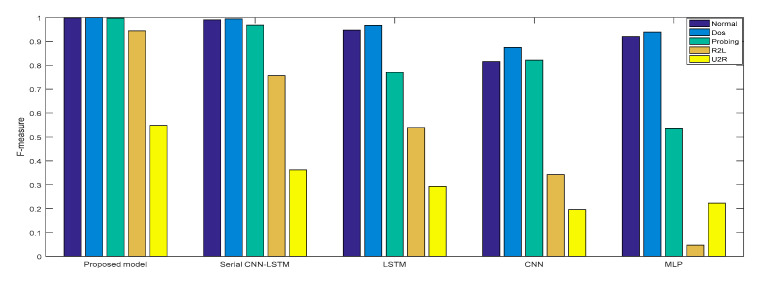
*F*-measure of NSL-KDD.

**Table 1 sensors-21-00626-t001:** Distribution of KDD Cup 99.

Type of Label	10%KDD Cup 99		Training Set		Test Set	
	Quantity	Ratio (%)	Quantity	Ratio (%)	Quantity	Ratio (%)
Normal	97,277	19.69	87,467	19.67	9813	19.86
Dos	391,458	79.24	352,405	79.26	39,053	79.06
Probing	4107	0.83	3685	0.83	422	0.85
R2L	1126	0.23	1018	0.23	108	0.22
U2R	52	0.01	45	0.01	7	0.01
Total	494,020	100	444,617	100	49,403	100

**Table 2 sensors-21-00626-t002:** Distribution of NSL-KDD.

Type of Label	NSL-KDD		Training Set		Test Set	
	Quantity	Ratio (%)	Quantity	Ratio (%)	Quantity	Ratio (%)
Normal	67,343	53.46	60,659	53.50	6684	53.05
Dos	45,927	36.46	41,323	36.45	4604	36.55
Probing	11,656	9.25	10,461	9.23	1195	9.49
R2L	995	0.79	884	0.78	111	0.88
U2R	52	0.04	48	0.04	4	0.03
Total	125,973	100	113,375	100	12,598	100

**Table 3 sensors-21-00626-t003:** Label numerical results.

Type of Label	Numerical Result
Normal	0
Dos	1
Probing	2
R2L	3
U2R	4

**Table 4 sensors-21-00626-t004:** Software and hardware configuration.

Project	Environment/Version
Operating System	Windows 10
CPU	i7-10700
Memory	32 G
GPU	GTX 2070 Super
Development Environment	Spyder3.0 (Python3.6)

**Table 5 sensors-21-00626-t005:** Setting of hyper-parameter.

Hyper-Parameter	Filter/Neurons
Conv + ReLU	8/16
LSTM hidden nodes	80
LSTM activation function	ReLU
Dense (Conv/LSTM)	128
Dense	256
Softmax	5
Cost function	Cross entropy
Batch size	128
Epoch	100

**Table 6 sensors-21-00626-t006:** Performance comparison.

Systems	KDD Cup 99			NSL-KDD		
	Accuracy (%)	*DR* (%)	*FPR* (%)	Accuracy (%)	*DR* (%)	*FPR* (%)
AE-CNN [32]	93.99	77.94	6.82	/	/	/
LSTM [36]	94.11	77.07	0.18	/	/	/
LSTM-RNN [37]	96.93	98.88	10.04	/	/	/
GA-ELM [35]	98.90	99.16	1.36	/	/	/
CNN-LSTM [41]	99.70	99.60	/	/	/	/
ELM [34]	98.94	98.37	0.72	97.58	97.69	2.22
ICNN [33]	/	/	/	95.36	96.99	0.76
CNN [31]	/	/	/	97.07	97.14	0.87
**Proposed**	**99.95**	**99.91**	**0.03**	**99.79**	**99.92**	**0.34**

## Data Availability

Publicly available datasets were analyzed in this study. This data can be found here: [http://kdd.ics.uci.edu/databases/kddcup99/kddcup99.html].

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
