# Peer review of "Intrusion Detection System in the Advanced Metering Infrastructure: A Cross-Layer Feature-Fusion CNN-LSTM-Based Approach"

_sensors, 2021, doi:10.3390/s21020626_

Round 1

Reviewer 1 Report

The paper presents an implementation of two deep learning techniques to be used on an IDS oriented to work into AMI.  The model was trained and tested using KDD-99 and NSL-KDD datasets.

Some points that need to be addressed by the authors are:

  • There is not clearly defined why the model proposed is optimized to be used into a AMI, in the form it is presented it could be used into any type of IP network.
  • Is important to justified the use of KDD-99 and NSL-KDD as datasets despite the criticized of them in the literature.
  • In Section 2, related work, need to include some works that already presented the aggregation of CNN and LSTM and specified why this work is different of them.
  • In Section 5.2 proposed the calculation of the Accuracy to evaluate the model. The literature criticized the use of the measure when unbalanced datasets are used as in your work.
  • More experimentations is suggested to validate the model, there is only presented one experiment that evaluate the performance.
  • Finally is not clear why this paper fits on the scope of this journal.

Author Response

Thank you very much for your valuable and instructive comments, attached please find our response letter.

Reviewer 2 Report

This paper presents an intrusion detection method using CNN and LSTM for AMI network. There are some major concerns about the paper:

  1. I don't think the title "cross-layer aggregation" is appropriate. The proposed scheme is basically a feature fusion that combines the features from CNN and LSTM. The inputs to CNN and LSTM are the same.
  2. It's not appropriate to call the features from LSTM "temporal features" since the input to LSTM is not time sequence data.
  3. I don't understand why CNN and LSTM combined can achieve significant better results than CNN and LSTM, especially R2L and U2R classes. Apparently CNN and LSTM can't detect those two classes but they can be mostly detected when CNN and LSTM combined which doesn't make sense.
  4. Cross validation (5-fold or 10-fold) should be used in evaluation.

There are some minor writing issues. For example. page 10, line 328, "up 99 dataset". 

Author Response

(The authors gave the same response as above.)

Round 2

Reviewer 2 Report

The authors have addressed majority of my comments. There are two further comments and suggestions:

(1) The authors show a serial CNN-LSTM architecture in the response. It is worthy to compare its performance with the proposed method to show the advantage of the proposed method.

(2) The author mention the datasets are imbalanced. Why not consider method dealing with imbalanced datasets such as upsampling or downsampling?

Author Response

(The authors gave the same response as above.)
